# Assigning Co-Regulated Human Genes and Regulatory Gene Clusters

**DOI:** 10.3390/cells10092395

**Published:** 2021-09-12

**Authors:** Tobias Strunz, Martin Kellner, Christina Kiel, Bernhard H. F. Weber

**Affiliations:** 1Institute of Human Genetics, University of Regensburg, 93053 Regensburg, Germany; Tobias.Strunz@klinik.uni-regensburg.de (T.S.); Martin.Kellner@klinik.uni-regensburg.de (M.K.); Christina.Kiel@klinik.uni-regensburg.de (C.K.); 2Institute of Clinical Human Genetics, University Hospital Regensburg, 93053 Regensburg, Germany

**Keywords:** expression quantitative trait loci, eQTL, colocalization, regulation of gene expression, co-regulation of gene expression

## Abstract

Elucidating the role of genetic variation in the regulation of gene expression is key to understanding the pathobiology of complex diseases which, in consequence, is crucial in devising targeted treatment options. Expression quantitative trait locus (eQTL) analysis correlates a genetic variant with the strength of gene expression, thus defining thousands of regulated genes in a multitude of human cell types and tissues. Some eQTL may not act independently of each other but instead may be regulated in a coordinated fashion by seemingly independent genetic variants. To address this issue, we combined the approaches of eQTL analysis and colocalization studies. Gene expression was determined in datasets comprising 49 tissues from the Genotype-Tissue Expression (GTEx) project. From about 33,000 regulated genes, over 14,000 were found to be co-regulated in pairs and were assembled across all tissues to almost 15,000 unique clusters containing up to nine regulated genes affected by the same eQTL signal. The distance of co-regulated eGenes was, on average, 112 kilobase pairs. Of 713 genes known to express clinical symptoms upon haploinsufficiency, 231 (32.4%) are part of at least one of the identified clusters. This calls for caution should treatment approaches aim at an upregulation of a haploinsufficient gene. In conclusion, we present an unbiased approach to identifying co-regulated genes in and across multiple tissues. Knowledge of such common effects is crucial to appreciate implications on biological pathways involved, specifically when a treatment option targets a co-regulated disease gene.

## 1. Introduction

In recent years, increasing attention has been given to the non-coding sequences of the human genome, highlighting the importance of common and rare genetic variants that influence disease etiology [1]. An immediate benefit is given for the evaluation of genome-wide association data as most of the genetic variants associated with complex traits are located in intronic or intergenic regions of the human genome [2,3]. In fact, many of these variants are considered to play a prominent role in the regulation of gene expression. Consequently, identifying and characterizing such variants in detail appears key to elucidating the biological mechanisms underlying the association signals [4].

Genome editing allows direct modifications of gene expression regulation using either transcriptional activators [5,6] or repressors [7]. These advances are of paramount importance for gene therapy approaches to human disease [8]. A first study of Matharu et al. (2019) [9] suggests that targeted activation of gene expression is suited to prevent clinical phenotypes attributed to haploinsufficiency. Such strategies, however, require a comprehensive understanding of the regulatory networks involved.

State of the art experimental approaches link regulatory regions to a gene of interest by chromosome conformation capture (CCC) techniques such as Hi-C [10,11]. These techniques are based on a fragmented library of cross-linked chromatin, which is analyzed with regard to DNA–DNA interactions using next generation sequencing [12]. To date, a number of CCC protocols have been developed, including ChIA-Pet, which combines Hi-C with chromatin immunoprecipitation to identify genomic regions interacting with a specific protein of interest [13]. ChIA-Pet enables the identification of regulatory networks under the condition that the corresponding transcription factor is known. Even though CCC techniques are highly sophisticated, the data generated often suffer from low resolution and cell-type specificity [14]. Consequently, experimental applications of the CCC methodology could be limited, although these approaches are explicitly suited to define topologically associating domains (TADs) in the genome [1].

An alternative approach defining regulatory sequences regardless of their genomic position correlates genetic variants with gene expression data. Such statistical concepts are generally based on linear models and lead to the discovery of expression quantitative trait loci (eQTL) [15]. Following the idea of eQTL, thousands of regulated genes were reported for single tissues like liver [16] or whole blood [17]. Furthermore, the Genotype-Tissue Expression (GTEx) consortium generated a gene expression regulation database, including 49 human tissues [18]. The occurrence of linkage disequilibrium (LD) in the human genome, however, frequently leads to the identification of multiple linked variants influencing the expression of the same gene [19,20]. Consequently, eQTL data on its own are mostly not sufficient to identify the causal regulatory element within an unknown DNA sequence. This limitation can, to some extent, be bypassed by colocalization analyses, which consider a complex genetic signal instead of focusing on a single genetic variant [21,22]. In this context, a complex genetic signal is defined as the entirety of linked variants that are associated with gene expression, and their defined relationship to each other. Colocalization tools, such as coloc [21] or eCaviar [23], have the power to identify co-regulated genes by comparing eQTL signals. Therefore, the combination of eQTL and colocalization studies is well suited to identify genes within the same TAD without the knowledge of the exact binding and interaction characteristics. Being informed about co-regulated genes could be helpful to further experimental designs, but is also helpful in evaluating potential therapeutic approaches affecting gene expression.

Here we present a comprehensive survey of co-regulated genes based on 49 tissues of the GTEx dataset. This highlights coregulation of gene expression as a common phenomenon and provides a basis to implement such an aspect into experimental approaches. An immediate medical relevance of our results is stressed for a situation where gene-therapy approaches aim to modify the expression of a co-regulated disease gene.

## 2. Materials and Methods

### 2.1. Genotype and Sample Quality Control

Whole genome sequencing (WGS) data of the GTEx Project (version 8, genome build hg38) [18] were retrieved from dbGaP (accession ID: phs000424.v8.p2) in VCF format [24]. Detailed information about genotype processing and quality control (QC) protocols is provided in detail elsewhere [18]. To determine ethnicity of samples, a principal component analysis (PCA) was carried out in R (version 3.3.1) [25] using the *snpgdsPC0A* [26] function based on 100,000 random genetic variants of each sample and the corresponding genotype information of the 1000 Genomes Project reference panel (Phase 3, release 20130502) [27]. The first two principal components (PCs) were plotted to determine ethnicity (Appendix A). Haplotype structures can vary between populations and, consequently, lead to different gene expression regulation mechanisms [28]. Therefore, we have manually chosen samples in direct proximity to the European (EUR) reference individuals in the genotype PCA (PC1 < −0.0078 and/or PC2 < −0.0125 in Appendix A). These samples represent the population with the highest sample size in GTEx. Altogether, genotypes from 694 out of 838 samples were included in the subsequent analysis. A minor allele frequency (MAF) threshold of 1% was applied resulting in 9,158,644 genetic variants. Handling of VCF data was performed using VCFtools (version 0.1.17) [24].

### 2.2. eQTL Calculation

Gene expression data from 49 tissues were downloaded via the GTEx Portal and filtered for EUR individuals based on the genotype PCA (Appendix A) [29]. The detailed data processing protocols are given elsewhere [18]. The sample sizes varied per tissue from 65 (KDNCTX) to 584 (MSCLSK) (Appendix A).

Local eQTL were calculated for each tissue separately based on linear regression models using *FastQTL* (version v2.184_gtex) as implemented in the GTEx version 8 analysis pipeline [18,30]. The mapping window to detect local eQTL was defined as 1 Mbp upstream and downstream of the transcription start site. Gender, the WGS platform, the WGS library construction protocol, the first five genotype PCs, and up to 60 PEER factors were included in the models as covariates. These covariates were provided by the GTEx Portal and are explained in [18]. *FastQTL* was applied in the *permute 1000 10000* mode to calculate beta distribution-extrapolated empirical *p*-values for each potential eGene. The *p*-values were then used to calculate gene level Q-values with the help of the false discovery rate approach as implemented in the R package *qvalue* (version 2.6.0) [31]. The lambda parameter was set to 0.85. A Q-value threshold of <0.05 was applied to identify genes regulated by at least one significant eVariant. To detect all significant eVariants, a tissue-specific genome-wide empirical *p*-value threshold was defined as described in [18]. This threshold was used to calculate a nominal *p*-value threshold for each gene based on the beta distribution parameters from *FastQTL*. Thereafter, *FastQTL* was run in normal mode to calculate nominal *p*-values of all local gene–variant associations. Nominal *p*-values from the linear regression model below the beforehand defined gene-specific threshold were considered significant.

### 2.3. Identification of Gene Expression Regulation Cluster

Regulatory clusters were defined as genomic regions containing multiple genes regulated by the same eQTL signal. This analysis required a two-step protocol to limit the computational and statistical burden.

First, significant eVariants were filtered for variants regulating two or more eGenes as only these genes have the potential to be regulated by the same genetic signal. By combining variants located on the same chromosome within an arbitrary distance of less than 2 Mbp, the variants were subsequently merged into genomic regions containing potentially co-regulated eGenes.

In a second step, all eGenes within those genomic regions were investigated for co-localization of their associated variants. This was performed using the *coloc.abf* function of the *coloc* package in R [21,22]. The method utilized the respective eQTL nominal *p*-values, effect sizes, variances of the effect sizes, as well as the SD of the expression for the investigated genes. In addition, the tissue-specific MAF was supplied for all variants as the sample size varied between tissues. *coloc.abf* further requires the specification of three informative prior probabilities. These are the probabilities that any random genetic variant in the region is associated with exactly gene one (p1), gene two (p2), or both genes (p12). The probabilities p1 and p2 were set to the gene-specific threshold of the corresponding eQTL analysis. The default for p12 was set to 5.0 × 10^−6^ as suggested by the author of the coloc package [22]. If either p1 or p2 was below 5.0 × 10^−6^, p12 was also adjusted to this smaller *p*-value. Next, the posterior probability that the same eQTL signal is regulating both genes (H4) was extracted from the co-localization analysis. If the analyzed gene pair did not share any overlapping variants within the 2 Mbp window, the H4 value was adjusted to 0. The H4 probabilities of all genes within one genomic region were then supplied to the *cliques* function of the igraph package (version 1.2.6) [32] in R to identify regulatory clusters. The H4 probability threshold for cluster definition was set to at least 0.8. Gene pairs within a single cluster fulfilling this criterion were defined as being regulated by the same eQTL signal. Visualization of exemplary clusters and results was performed with the help of the ggplot2 package (version 3.2.1) in R [33].

### 2.4. Collection of Haploinsufficiency Genes

Two different resources were exploited to generate a list of genes which are known to cause haploinsufficiency phenotypes. The ClinGen database [34] provides a list of genes causing potential dosage sensitivity phenotypes and assigns them to different categories [35]. A list of 312 genes with “sufficient evidence for haploinsufficiency” was downloaded from the ClinGen database website [36]. Furthermore, Matharu and colleagues [9] published a list of 660 genes leading to haploinsufficiency disease (see “Appendix A” in [9]). Combining the two resources led to a list of 713 unique genes with evidence for causing haploinsufficiency-related phenotypes (Appendix A).

## 3. Results

### 3.1. Identification of Co-Regulated Genes

Our study aimed to identify co-regulated genes and their organization in gene expression regulation clusters. To this end, we designed a workflow starting with the calculation of local eQTL. We then filtered for all significant variants (eVariants) regulating the expression of two or more genes (eGenes) as only these genes have the potential to be regulated by the same genetic signal. A subsequent pair-wise colocalization analysis revealed co-regulated eGenes and facilitated the identification of genes clustered due to underlying genetic signals. The following characterization of the identified regulatory clusters was intended to dissect the results at specified levels to include gene-centered, cluster-centered, and tissue-centered perspectives as well as an example highlighting medical consequences of the data presented (Figure 1).

Exploiting available data from the GTEx project, eQTL were calculated for 49 different tissues based on gene expression, genotype, and covariate data of 694 samples from donors of European descent (Appendix A). For each tissue, the sample size varied widely from 65 (kidney cortex, KDNCTX) to 584 (muscle skeletal, MSCLSK) (Appendix A). Our analysis included 39,832 expressed genes of which 33,488 (84,1%) were genetically regulated in at least one tissue (Appendix A). Remarkably, 14,636 (43.7%) of the latter gene group showed a significant colocalization (coloc probability ≥ 0.80) with at least one other gene, generating a list of 49,637 co-regulated gene pairs (Appendix A) across the 49 tissues. A closer look at the underlying eQTL revealed that 37,200 (74.9%) of the 49,637 gene pairs were simultaneously up- or downregulated. The remaining 12,437 (25.1%) gene pairs were regulated by the same genetic signals but in opposite directions.

Of the 49,637 gene pairs, we identified a total of 16,673 unique combinations across multiple tissues (Appendix A). Determining the genomic position of the co-regulated genes revealed that in 86.2% (14,368/16,673) the gene loci did not physically overlap (Appendix A). Their mean distance was 122,265 bp (standard deviation, SD: 267,498 bp) with 885 colocalizing genes having a physical distance of more than 1 Mbp (Appendix A). The remaining co-regulated genes were either partially (4.5%, 759/16,673) or fully overlapping (9.3%, 1546/16,673). Furthermore, we observed that 41.8% (6977/16,673) of the co-regulated gene pairs were identified in multiple tissues with 2469 gene pairs being found in five or more tissues (Appendix A). Remarkably, one gene pair—*FAM157C*/*RP11-356C4.5*—was expressed in all tissues analyzed and showed a significant co-regulation in 45 of the 49 tissues (not significant for tissues LUNG, SPLEEN, TESTIS, WHLBLD (whole blood)).

While the number of significant eGenes was highly correlated with the sample size of the respective tissue (*p*-value = 1.6 × 10^−15^, R^2^ = 0.744) (Figure 2A), this was only partially the case for co-regulated genes. For tissues comprising less than 200 samples, there was a strong correlation of the sample size with the number of co-regulated genes (*p*-value = 4.2 × 10^−8^, R^2^ = 0.736), whereas no correlation was found above a sample size of 200 (*p*-value = 0.6, R^2^ = 0.010) (Figure 2B). We therefore considered the 24 tissues with a sample size above 200 (Appendix A and Figure 2C) for our subsequent evaluations, as the findings in these tissues appear not to be biased by the underlying sample size. Still, Appendix A summarize the findings for all tissues investigated regardless of sample size to facilitate an overall evaluation of tissue-specific effects.

### 3.2. Gene Expression Regulation Cluster

We grouped the colocalizing genes into gene expression clusters that were regulated by the same genetic signal. Altogether, 10,482 unique clusters were identified in the 24 tissues with a sample size above 200 (Appendix A). Overall, the cluster size varied widely including two to a maximum of nine genes (Figure 3A). The two largest clusters were found in colon transverse (CLNTRN) tissue and comprised the genes *TMEM141*, *CCDC183*, *CCDC183-AS1*, *MAMDC4*, *LCN12*, *ABCA2*, *NPDC1*, *ENTPD2*, and additionally either *PHPT1* or *RABL6*. Furthermore, many clusters were detected in multiple tissues and sometimes included additional genes. For example, the *FLG*/*HRNR* cluster in adipose subcutaneous (ADPSBQ) tissue was additionally co-regulated together with the antisense transcript *FLG-AS1* in another 18 tissues (Appendix A). Nevertheless, the overall distribution of cluster sizes was highly comparable between tissues (Figure 3B), with a cluster size of two coregulated genes being the most prominent finding (81.2–88.6% of all clusters).

The number of clusters per tissue ranged from 645 (cells cultured fibroblasts, FIBRBLS) to 1243 in testis (TESTIS) tissue. A comparison between tissues revealed that each tissue pair has at least 85 gene expression regulation clusters in common (TESTIS and WHLBLD, Figure 4). The mean number of shared clusters between tissues was 212.4 (SD: 66.2). Furthermore, some highly similar tissues—such as skin not sun exposed suprapubic (SKINNS) and skin sun exposed lower leg (SKINS)—share up to 439 clusters, which account for approximately 50% of all clusters found in the respective tissues. Remarkably, four tissues showed less shared regulation clusters in comparison to all other tissues (mean: 227.0, SD: 62.2) (Figure 4): TESTIS (mean: 135.1, SD: 21.4), WHLBLD (mean: 121.1, SD: 23.2), FIBRBLS (mean: 148.0, SD: 25.6), and MSCLSK (mean: 154.1, SD: 27.4).

### 3.3. Potential Clinical Consequences for Genes within Expression Regulation Clusters

Gene expression regulation cluster may have an impact on clinical applications such as defined gene therapy approaches. For example, a recent study published by Matharu et al. (2019) applied CRISPR-mediated activation (CRISPRa) to enhance gene expression aiming to ameliorate a phenotype caused by disease-associated haploinsufficiency [9]. Such an approach could lead to off-target gene expression effects if the target gene is part of a gene expression regulation cluster (Figure 5A).

To appreciate the relevance of such a situation, we generated a list of 713 genes known to cause phenotypes due to haploinsufficiency (Appendix A) and tested their occurrence in the gene expression regulation clusters across all 49 tissues. Remarkably, 231 of the 713 genes (32.4%) are part of at least one cluster, affecting 429 different clusters altogether (Figure 5B, Appendix A). Besides the genes causing haploinsufficiency, these clusters include an additional 414 potential off-target genes. The 429 affected clusters were distributed across the genome. Furthermore, nearly all of the 49 tissues investigated—except KDNCTX, which had the smallest sample size of all tissues (*n* = 65)—included at least one cluster harboring a haploinsufficiency gene.

## 4. Discussion

Here, we report on the co-regulation of gene expression based on comprehensive datasets from 49 tissues that were extracted from the GTEx project [18]. We show that 14,636 of 33,488 genetically regulated genes (43.7%) are co-regulated together with at least one other gene, resulting in 14,727 unique expression clusters across the tissues analyzed. In these clusters, up to nine genes are co-regulated by the same genetic signal, while pairwise tissue comparison highlights that each tissue pair shares, on average, 212 clusters. Since co-regulation is such a common phenomenon, we provide an exemplary scenario considering the field of innovative therapeutics where modifying gene expression within a gene regulation cluster could have significant repercussions on the treatment outcome, or more specifically on possible side-effects, and thus need to be acknowledged with great care.

The mechanisms underlying the regulation of local gene expression have been extensively studied although for individual genes only [1,37]. Such studies show that genetic variation can affect promoters [38], enhancers [39], and silencers [40], but also intronic elements such as splice site consensus sequences and consequently the correct splicing of transcripts [18]. At the chromatin level, genetic variation may alter TAD boundaries [41], potentially leading to changes in the regulation of multiple genes [42,43]. Undirected large-scale approaches, such as eQTL analyses, can be complicated by complex LD structures. CCC methods, on the other hand, are successful in revealing direct interactions of DNA elements [12,14] and identify TADs [44]. Still, the knowledge of interaction sites per se is often less suited for identifying regulated genes, as the resolution is generally low having to deal with gene loci that position genes in close proximity to each other [14]. Our database of co-regulated genes generated as a result of this study is based on eQTL and colocalization analyses and offers a new perspective to compensate for the shortcomings of existing approaches by pointing directly to the gene of interest. It is therefore recommended to integrate our data directly in the result interpretation of published CCC studies as this will greatly enhance the value of both approaches. Given that CCC data are often generated in a cell-specific context, it is advisable to perform the comparison specifically in a context of a defined research question.

Regarding advantages and limitations of the database compiled in this study, we mainly need to consider two aspects. Firstly, while genes can be regulated by multiple genetic signals, the number of independent signals is strongly correlated with the associated sample size [18]. We therefore only considered the most prominent signal for each gene to not further complicate the comparison of tissues and to avoid an unfavorable dependency on sample size. While this procedure could be prone to miss some of the interactions, the ones we identified can be viewed with high confidence, specifically as they are based on the strongest eQTL signal for each locus [22]. Secondly, most eQTL studies are performed in bulk tissue, which can obviously result in unidentified eQTL by a dilution effect, as it is known that gene expression regulation is frequently a cell type-specific process [45,46]. However, our findings that highlight shared gene expression regulation clusters between tissues point to tissue overlapping mechanisms, even if our database allows no direct conclusion about cell type specificity.

Knowledge about clusters of gene expression regulation should be transferable to generate novel hypotheses and experimental designs. While each cluster needs to be considered individually, a number of mechanisms underlying the co-regulation are conceivable. For one, the effect direction of gene expression regulation provides useful information about the potential reason why genes are co-regulated. Our study revealed that 37,200 of the 49,637 co-regulated gene pairs (74.9%) are regulated in the same direction which brings us to hypothesize that the respective genes could play a direct or indirect role in the same pathway. A prominent example for the coordinated activation of genes are the *HOX* gene clusters, for which expression was studied extensively in the context of embryogenesis [47,48]. Of note, some of the largest clusters in our study were identified on chromosome two and seven containing up to eight genes of the *HOXD* and *HOXA* gene family, respectively. Although our results are based on differentiated tissue cells, the concordance with previously published studies shows the validity of our approach. Of note, the remaining 12,437 of the 49,637 co-regulated gene pairs (25.1%)—which are regulated in opposite directions—also have the potential to directly point to underlying biological mechanisms. A prominent example here is the gene expression regulation cluster including *ATE1* and *ATE1-AS1*, in which simultaneous downregulation of the antisense transcript *ATE1-AS1* is associated with increased expression of *ATE1*. This effect can be observed in various tissues and previous studies have demonstrated that *ATE1* is regulated by a bidirectional promoter [49]. Kalinina et al. (2021) suggested that competing RNA structures at this locus affect splicing of the *ATE1* gene [50]. These examples demonstrate that our results are in line with published studies, which were performed using different methods and approaches.

A significant strength of our dataset lies in the potential to evaluate the impact of experimental approaches targeting a DNA sequence or the regulation of transcription. For example, the gene *FLG* encodes profilaggrin whose loss is known to cause impaired keratinization [51]. A hypothetical gene therapy to prevent this phenotype could use the CRIPSR/Cas technology to facilitate an upregulation of *FLG* expression. Based on our findings, *FLG* is co-regulated with *HRNR* although in opposite direction. Thus, manipulating *FLG* expression may consequently alter *HRNR* expression, whereby it should be noted that the latter protein plays a role in atopic dermatitis [52]. In this scenario, *HRNR* needs to be considered as a potential off-target, although classical off-target prediction tools are based on sequence alignment algorithms and do not consider gene expression co-regulation [53,54]. The high proportion of 32.4% (231) of 713 haploinsufficiency genes that are in a regulation cluster illustrates that this phenomenon is in fact common and should be anticipated in any gene therapy designs.

## 5. Conclusions

Here, we present our findings in a novel database providing access to thousands of co-regulated genes in 49 tissues. Co-regulated genes are in close proximity or at distances of up to 2 Mbp and many clusters of expression regulation are shared between tissues. We highlight the prevalence of gene expression co-regulation and provide an outset for further follow-up studies. Finally, our results have a strong impact on current and future therapeutic approaches aiming at the modification of gene expression. Specifically, this is demonstrated for gene therapies targeting genes causing disease due to haploinsufficiency. Co-regulated genes are likely affected by such a treatment possibly triggering severe side effects.

## Figures and Tables

**Figure 1 cells-10-02395-f001:**
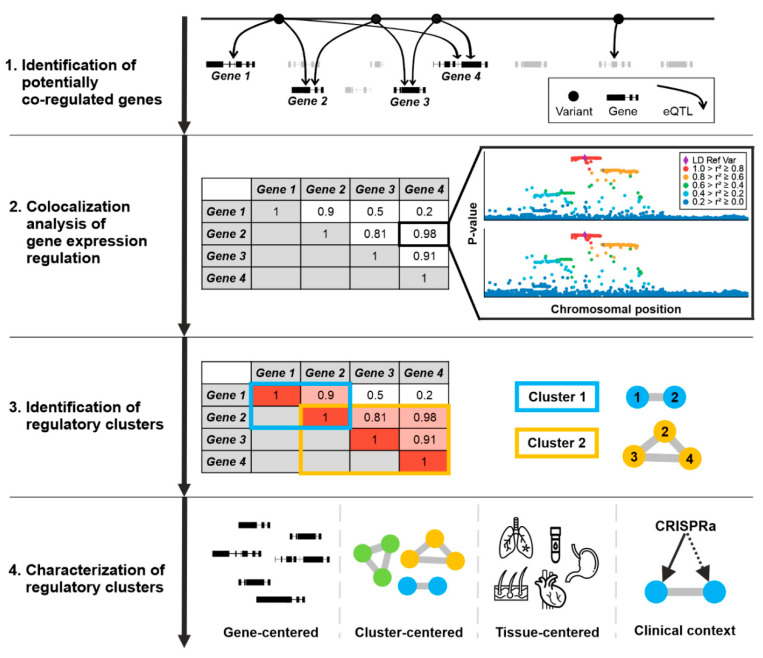
Schematic workflow of the gene expression regulation cluster analysis. Clusters of co-regulated gene expression were defined as genomic regions containing multiple genes regulated by the same eQTL signal. In a first step, the eQTL results were filtered for eVariants regulating at least two eGenes. The corresponding eQTL signals were then analyzed for colocalization (Step 2). Gene pairs with a colocalization probability of 0.80 and higher were considered to be co-regulated. Next, eGenes with colocalizing eQTL signals were grouped in regulatory clusters (Step 3). These clusters were subsequently investigated on multiple levels considering a gene-centered, a cluster-centered, and a tissue-centered perspective to enable a comprehensive interpretation of results. Finally, genes with clinical relevance to haploinsufficiency-related disease were investigated for their involvement in gene expression regulation clusters and the impact of potential therapeutic approaches. For example, the CRISPRa technology [9] could affect gene expression of the targeted gene but also of off-target genes within the same cluster.

**Figure 2 cells-10-02395-f002:**
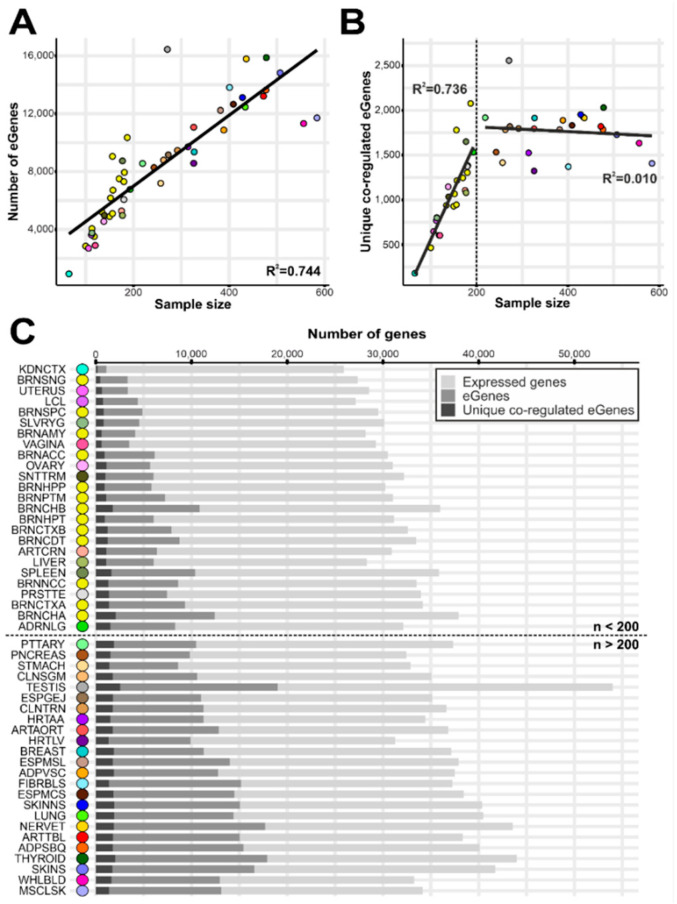
Gene-centered analysis of co-regulated genes. eQTL were calculated in 49 tissues from GTEx project data based on gene expression and genotype data of 694 samples from donors of European descent. The sample sizes varied widely between the tissues, namely from 65 (KDNCTX) to 584 (MSCLSK) (Appendix A). (**A**) Number of significant eGenes as a function of sample size. The regression coefficient (R^2^) of the linear model is 0.744 (*p*-value = 1.6 × 10^−15^) (**B**) Number of significant eGenes, co-regulated with at least one other eGene, as a function of sample size of the respective tissue. A significant correlation was observed for tissues with less than 200 samples (*p*-value = 4.2 × 10^−8^, R^2^ = 0.736), whereas the linear regression model was not significant considering tissues with samples sizes above 200 (*p*-value = 0.6, R^2^ = 0.010). Consequently, the number of eGenes over 200 samples per tissue with at least one co-regulation partner is not correlated with sample size. (**C**) Visualization of the number of expressed genes (light grey), the number of eGenes (medium grey), and the number of eGenes, which are co-regulated with at least one other eGene (dark grey) in the 49 tissues analyzed. The tissues are ordered from top to bottom by sample size in decreasing order. The black dotted line separates tissues with 200 or less samples (above) from those with sample sizes over 200 per tissue (below). The color code was retrieved from the GTEx project [18]. A detailed list of tissues analyzed and their respective abbreviation is given in Appendix A.

**Figure 3 cells-10-02395-f003:**
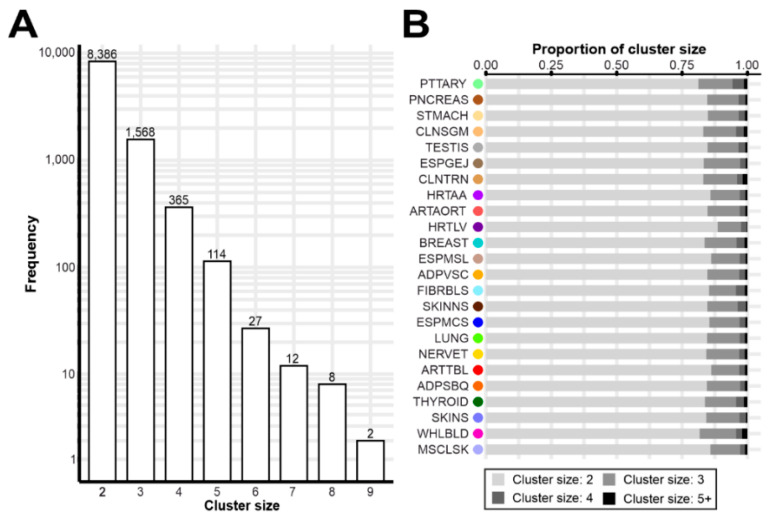
Cluster sizes and distribution. (**A**) Histogram representing the sizes of clusters of gene expression regulation by logarithmic scale. Only tissues with a sample size above 200 were considered and clusters, which were identified identically in multiple tissues were included only once. (**B**) Relative distribution of cluster sizes within tissues. The grey scale represents the size ranging from 2 (light grey) to above 5 (5+, dark grey). A detailed list of tissues analyzed, and their respective abbreviations, is given in Appendix A.

**Figure 4 cells-10-02395-f004:**
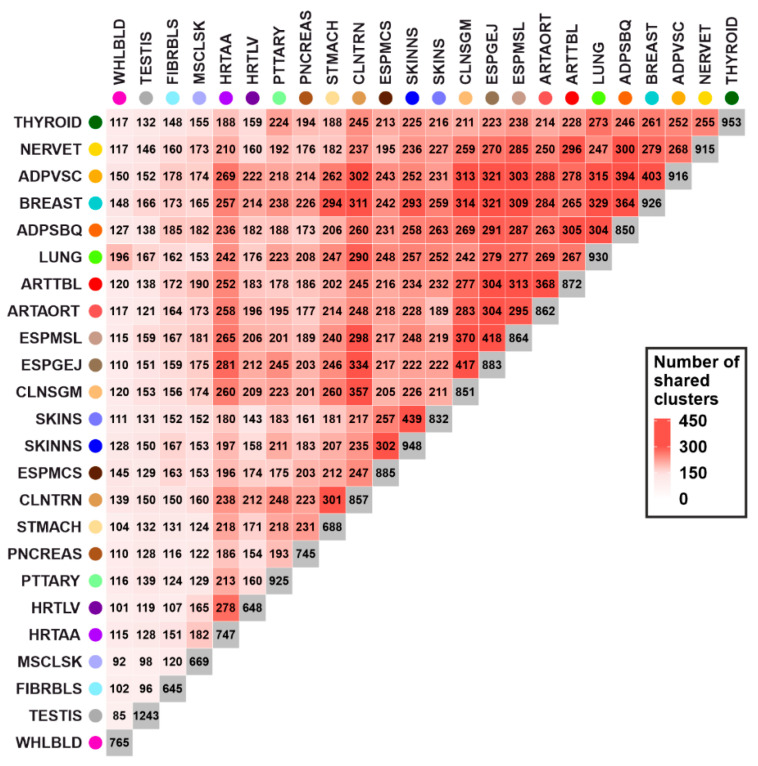
Comparison of gene expression regulation clusters between tissues. The heatmap displays the number of identical clusters between tissue pairs. Clusters in one tissue, which included additional genes in the other tissue, were also counted in this comparison. The color scale represents the number of shared clusters between tissues. Hierarchical clustering of tissues was performed based on commonly regulated gene expression clusters. Only tissues with a sample size above 200 were included in this analysis. A detailed list of tissues analyzed, and their respective abbreviations, is given in Appendix A.

**Figure 5 cells-10-02395-f005:**
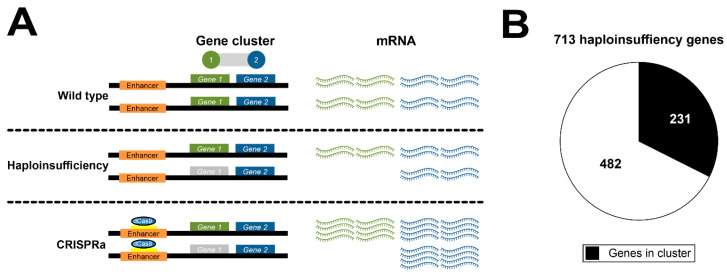
The potential role of gene expression regulation cluster in gene therapy. (**A**) Schematic overview of a gene therapy approach modifying expression of a gene, whose disease-associated deletion causes a haploinsufficiency phenotype. In the regular situation, gene 1 (green) and gene 2 (blue) are expressed but are known to be regulated by the same genetic signal (gene cluster). The disease-associated deletion of one copy of gene 1 is leading to haploinsufficiency. A planned gene therapy—e.g., CRISPR/Cas-mediated activation (CRIPSRa)—is intended to enhance expression of the regular copy of gene 1 to ameliorate the disease phenotype. This procedure potentially also increases gene 2 expression as the latter gene is part of a common regulatory gene cluster with gene 1. Gene 2 is therefore a potential off-target, which is currently not considered by common off-target prediction approaches. (**B**) Occurrence of haploinsufficiency genes in gene expression regulation clusters as defined in this study. Of 713 genes known for causing haploinsufficiency-related phenotypes, 231 (32.4%) were found to be part of gene expression regulation clusters.

## Data Availability

Genotype, gene expression, and covariate data of the GTEx project are available in dbGaP (accession ID: phs000424.v8.p2) or the GTEx portal (www.gtexportal.org). All data generated or analyzed during this study are included in this published article and its Appendix A.

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
