# Peer review of "Assigning Co-Regulated Human Genes and Regulatory Gene Clusters"

_cells, 2021, doi:10.3390/cells10092395_

Round 1
Reviewer 1 Report
In thi very interesting and novelty report, Strunz et al., describe the identificaction of co-regulated eGenes arranged in gene expression clusters associated to several eQTL by analyzing transcriptome and genotype data of 49 tissues derived from 694 samples from donors of European descent of GTEx project.
This work lead to the identification of several regulatory clusters (defined as genomic regions containing multiple genes regulated by the same eQTL signal), could aid to design a rationale of gene therapeutic approaches. Of special interest, the authors identified a high proportion of haploinsufficiency genes (32.4%) into regulation clusters, which in a near future, could be used to design in a rationale way, the correction of severa human diseases related to haploinsufficieny mechanisms (i.e. CRISP/Cas9 upregulation) and at the same time, avoiding adverse effects by the knowledge of co-regulation (i.e. , FLG co-regulation with HRNR). Interestingly the authors identified that co-regulated genes are in close proximity or at distances of up to 2 Mbp and many clusters of expression regulation are shared between tissues. Also, the authors describe punctually the possible limitations of their study.
I have only two minor suggestions:
- Please define and briefly explain the cited Hi-C approach (lane 47).
- Please consider to include into discussion section, instead of results one, the point 3.3 (Relevance of gene expression regulation cluster for gene therapy approaches).
Author Response
REVIEWER #1
Reviewer: In this very interesting and novelty report, Strunz et al., describe the identification of co-regulated eGenes arranged in gene expression clusters associated to several eQTL by analyzing transcriptome and genotype data of 49 tissues derived from 694 samples from donors of European descent of GTEx project. This work lead to the identification of several regulatory clusters (defined as genomic regions containing multiple genes regulated by the same eQTL signal), could aid to design a rationale of gene therapeutic approaches. Of special interest, the authors identified a high proportion of haploinsufficiency genes (32.4%) into regulation clusters, which in a near future, could be used to design in a rationale way, the correction of several human diseases related to haploinsufficiency mechanisms (i.e. CRISP/Cas9 upregulation) and at the same time, avoiding adverse effects by the knowledge of co-regulation (i.e. , FLG co-regulation with HRNR). Interestingly the authors identified that co-regulated genes are in close proximity or at distances of up to 2 Mbp and many clusters of expression regulation are shared between tissues. Also, the authors describe punctually the possible limitations of their study.
Answer: We greatly appreciate the reviewer’s highly positive assessment of the manuscript and the accuracy of highlighting the major points of our work. We also want to thank the reviewer for his/her valuable time to make suggestions for improving the manuscript.
Reviewer: I have only two minor suggestions:
Please define and briefly explain the cited Hi-C approach (lane 47).
Answer: We have now added additional information to the underlying principles of Hi-C (please see lines 49-56)
Reviewer: Please consider to include into discussion section, instead of results one, the point 3.3 (Relevance of gene expression regulation cluster for gene therapy approaches).
Answer: We appreciate the suggestion to transfer our analyses on the potential clinical consequences of gene expression regulation cluster into the discussion section. The objective of our study was to describe and to better understand the phenomenon of gene expression co-regulation by including a gene-centered, a cluster-centered, and a tissue-centered perspective. The transfer of section 3.3 (Potential clinical application of gene expression regulation cluster) into the discussion would shift our focus very much on the clinical side, putting the other perspectives into the background. For this reason, we would prefer to retain the structure of the manuscript to allow for a multi-layered interpretation of the results.

Reviewer 2 Report
The work presents extensive analysis of gene expression regulation region in human genome using eQTL data. Clustering of regulatory regions by gene expression in tissues and eQTL is novel approach.
I'd suggest fix some moment in the text. First need to update the Abstract to avoid exact numbers, just for convenient reading. Try to avoid abbreviations (eGenes - is redundant new abbreviation, used only once in the Abstract.) Instead of 'Of 33,488 eGenes' may write 'From about 33 thousands regulated genes...'. The abbreviations and exact numbers could be given in the main paper text.
In addition this number contradict to number of genes given in line 296 (in Discussion)
Avoid bulk citations like [5–7] (three ore more together). Please rephrase to show which reference is for genome editing, which is for disease case.
Line 46: 'chromosome conformation capture (CCC) techniques such as Hi-C'
I think this point could be commented in more details. There are several Hi-C variants including ChIA-PET method related to transcription factor binding and clusters detection data. There are open data on mapped TAD - chromosome domains that could be compared to the clusters found in this paper.
Line 56: 'consortium and others' -
- please change the citation - who is 'other', name the sources.
Line 85: 'Only samples clustering next to the European' -
need rephrase. Why Europeans only? It is explained later in the text, but need comment at first mention in the text.
What means 'clustering next to'? Need comment what is 'next' - some proximity by which measure? In the plot only?
Line 146: 'On January 15 th, 2021, a list of 312 genes...'
change the phrase, make it as reference, name the source. The data (15.01.21) could be given in the reference as the access date, not in the text.
Line 148: Maharu et al. - put the reference in brackets in the text after the name.
Line 264: Section '3.3 Relevance of gene expression regulation cluster for gene therapy approaches' looks redundant since there is only Figure 5, and no discussion about gene therapy.
I'd recommend rename this subsection and keep the Figure. Or add some text describing gene therapy with references to this section.
Lines 394-397 - text repeat. Even if it is formally correct, try to rephrase. It has no sense to repeat.
Author Response
Reviewer 2
Reviewer: The work presents extensive analysis of gene expression regulation region in human genome using eQTL data. Clustering of regulatory regions by gene expression in tissues and eQTL is novel approach. I'd suggest fix some moment in the text. First need to update the Abstract to avoid exact numbers, just for convenient reading. Try to avoid abbreviations (eGenes - is redundant new abbreviation, used only once in the Abstract.) Instead of 'Of 33,488 eGenes' may write 'From about 33 thousands regulated genes...'. The abbreviations and exact numbers could be given in the main paper text.
Answer: We appreciate the reviewer’s suggestions and have amended the Abstract accordingly.
Reviewer: In addition, this number contradict to number of genes given in line 296 (in Discussion)
Answer: We thank the reviewer for pointing out our oversight. Indeed, we named the number of co-regulated genes once in the context of all expressed genes and thereafter in the context of all regulated genes. We have now reworded the respective discussion section to avoid confusions (line 310).
Reviewer: Avoid bulk citations like [5–7] (three ore more together). Please rephrase to show which reference is for genome editing, which is for disease case.
Answer: We have now rephrased the corresponding text passages and have resolved the bulk citations (for example, see lines 42-44; 63-66; 326-327).
Reviewer: Line 46: 'chromosome conformation capture (CCC) techniques such as Hi-C'. I think this point could be commented in more details. There are several Hi-C variants including ChIA-PET method related to transcription factor binding and clusters detection data. There are open data on mapped TAD - chromosome domains that could be compared to the clusters found in this paper.
Answer: We thank the reviewer for this valuable suggestion and have added further explanations about chromosome conformation capture (CCC) in our introduction section (lines 49-56). Specifically, we have now explained that other CCC methods exist, which facilitate investigations into various research areas. For example, ChIA-PET has the potential to identify regulatory networks, particularly if the transcription factor of interest is already known. We fully agree that a comparison of our results with current CCC databases greatly increases the value of both resources. As CCC data are generated in a cell-specific manner, a general comparison with eQTL results from bulk tissue is only possible to a limited extent. Therefore, the comparison should be performed in follow-up studies that investigate more specific questions. We have included this aspect in our discussion to simplify the application of our database (lines 332-336).
Reviewer: Line 56: 'consortium and others' - please change the citation - who is 'other', name the sources.
Answer: Done.
Reviewer: Line 85: 'Only samples clustering next to the European' - need rephrase. Why Europeans only? It is explained later in the text, but need comment at first mention in the text. What means 'clustering next to'? Need comment what is 'next' - some proximity by which measure? In the plot only?
Answer: We apologize for our unclear writing and have amended the paragraph accordingly to add further explanations about the criteria for sample inclusion. Principally, our analysis was based on European samples only, as they represent the largest sample size in GTEx. Importantly, it was shown by Gay and colleagues (doi:10.1186/s13059-020-02113-0) that gene expression regulation differs between populations. We selected the samples manually by plotting the two principal components of the genotype principal component analysis. Samples next to the European samples of the 1000 Genomes Project reference panel (PC1 < -0.0078 and/or PC2 < -0.0125 in Figure S1) were kept for analysis (lines 95-99).
Reviewer: Line 146: 'On January 15 th, 2021, a list of 312 genes...'. Change the phrase, make it as reference, name the source. The data (15.01.21) could be given in the reference as the access date, not in the text.
Answer: Done.
Reviewer: Line 148: Maharu et al. - put the reference in brackets in the text after the name.
Answer: Done.
Reviewer: Line 264: Section '3.3 Relevance of gene expression regulation cluster for gene therapy approaches' looks redundant since there is only Figure 5, and no discussion about gene therapy. I'd recommend rename this subsection and keep the Figure. Or add some text describing gene therapy with references to this section.
Answer: We agree with the reviewer that the name of the subsection was slightly misleading. For this reason, we have changed it to “Potential clinical consequences for genes within expression regulation clusters” and have kept the rest of the paragraph as suggested.
Reviewer: Lines 394-397 - text repeat. Even if it is formally correct, try to rephrase. It has no sense to repeat.
Answer: Done.
